# Virtual Reality Technology to Enhance Conventional Rehabilitation Program: Results of a Single-Blind, Randomized, Controlled Pilot Study in Patients with Global Developmental Delay

**DOI:** 10.3390/jcm12154962

**Published:** 2023-07-28

**Authors:** Carmela Settimo, Maria Cristina De Cola, Erica Pironti, Rosalia Muratore, Fabio Mauro Giambò, Angelo Alito, Maria Tresoldi, Margherita La Fauci, Carmela De Domenico, Emanuela Tripodi, Caterina Impallomeni, Angelo Quartarone, Francesca Cucinotta

**Affiliations:** 1IRCCS Centro Neurolesi Bonino Pulejo, 98124 Messina, Italy; mariacristina.decola@irccsme.it (M.C.D.C.); rosalia.muratore@irccsme.it (R.M.); fabio.giambo@irccsme.it (F.M.G.); maria.tresoldi@irccsme.it (M.T.); margherita.lafauci@irccsme.it (M.L.F.); carmela.dedomenico@irccsme.it (C.D.D.); emanuela.tripodi@irccsme.it (E.T.); caterina.impallomeni@irccsme.it (C.I.); angelo.quartarone@irccsme.it (A.Q.); francesca.cucinotta@irccsme.it (F.C.); 2Woman-Child Department, Unit of Child Neurology and Psychiatry, Policlinico Riuniti Foggia, 71122 Foggia, Italy; erica.pironti@gmail.com; 3Department of Biomedical, Dental Sciences and Morphological and Functional Images, University of Messina, 98122 Messina, Italy; alitoa@unime.it

**Keywords:** neurodevelopmental disorders, virtual reality exposure therapy, child, preschool, randomized controlled trial, early intervention, educational

## Abstract

Global developmental delay (GDD) is a complex disorder that requires multimodal treatment involving different developmental skills. The objective of this single-blind, randomized, controlled pilot study is to evaluate the feasibility and effectiveness of conventional rehabilitation programs integrated with the BTs-Nirvana virtual reality system. Patients with GDD aged 12 to 66 months were enrolled and treated for a 48-session cycle. Patients were randomized into two groups, (1) conventional treatment and (2) conventional treatment supplemented with the use of BTs-Nirvana, in a 1:1 ratio. Before and after treatments, areas of global development were tested with the Griffiths-III Mental Developmental Scale and the clinical indicator of global improvement were measured with the Clinical Global Impressions-Improvement (CGI-I). Feasibility was confirmed by the high retention rate. The experimental group presented a significantly improvement in General Quotient (GQ) after treatment (GQ, *p* = 0.02), and the effect of the two treatments was significantly different in both the GQ (t =2.44; *p* = 0.02) and the Foundations of Learning subscale (t =3.66; *p* < 0.01). The overall improvement was also confirmed by the CGI-I (*p* = 0.03). According to these preliminary data, virtual reality can be considered a useful complementary tool to boost the effectiveness of conventional therapy in children with GDD.

## 1. Introduction

Global Developmental Delay (GDD) is a common neurodevelopmental disorder which affects 1–3% of children aged 5 years or younger [1]. It is characterized by deficits in developmental milestones in several areas of intellectual functioning [2]. GDDs are usually identified by caregivers, by teachers at school who raise concerns, or during routine clinical evaluations by the pediatrician [3]. It can be caused by specific conditions which are not always easy to identify. Chromosomal abnormalities, perinatal asphyxia, preterm birth, cerebral dysgenesis, psychosocial deprivation, and toxin exposure are some of the possible causes [4].

Recent data from the literature have underlined the importance of early diagnosis followed by appropriate therapeutic management. In fact, GDD could evolve into different neurodevelopmental disorders, especially if associated with other risk factors, such as intra-uterine growth retardation, nutrient deficiencies, breastfeeding and maternal education, scarce social and economic conditions, poor learning opportunities, inadequate quality of caregiver–child interactions [5]. The recognition of risk factors and early treatments can significantly influence the long-term outcome of developmental disability [6]. The importance of continuously stimulating children is well known, especially during the earliest stages of development. In fact, the young nervous system is capable of producing numerous new behaviors to interact with the environment and adapting to it [7].

For patients affected by GDD, the Italian National Health Service provides “speech therapy” and “neuropsychomotor therapy”. “Neuropsychomotor therapy” is a typical Italian rehabilitation approach for patients from birth until 18 years old, affected from neurodevelopmental disorders. It is similar to “play therapy” and “developmental therapy” practiced in other countries [8]. This therapy aims to strengthen motor, functional, affective, relational, and cognitive areas by trying to stimulate active learning through toys and interactive games; therefore, the ultimate goal of this kind of therapy is the harmonious integration of all functional areas during the growth process through a comprehensive approach [9]. Speech and neuropsychomotor therapy are usually prescribed to child and adolescent by psychiatrists or neurologists and delivered by community child neuropsychiatry services.

In recent decades, the use of virtual reality (VR) in rehabilitation has become more and more popular for its possible implementation in innovative treatments in cognitive-motor domain. Many studies suggest that VR can constitute a motivating and fun rehabilitation approach, being more engaging than conventional therapy or educational programs, both for adults and children [10,11]. This system allows naturalistic behaviors to be enacted in a controlled environment and permits therapists to adjust multimodal stimulus according to patients’ characteristics and needs [12]. VR-based rehabilitation promotes implicit learning, offering repetitive and intensive tasks with immediate sensorimotor feedback. Virtual reality, being associated with the idea of playing, allows the children an unconscious learning process [13].

In recent literature, several studies have been conducted to test the effectiveness and feasibility of virtual reality system for rehabilitative treatment in various neurodevelopmental disorders. Among these, probably one of the disorders in which this hypothesis has been most tested is infant cerebral palsy, in which gross motor function appears to improve significantly, although the effect on the daily living ability remains controversial [14]. Several reviews on autism spectrum disorder described the usefulness of VR tools for children affected by this heterogeneous neurodevelopmental disorder, particularly on social and emotional skills training [15,16], and as support for social situations that can be generalized in real-world [17]. Moreover, 83% of the articles reviewed by Goharinejad et colleagues [18] described the benefits of virtual, augmented, and mixed reality technologies to ameliorate symptoms of attention deficit hyperactivity disorder.

Among the high-tech tools used in recent years for neuropsychomotor rehabilitation, the BTs-Nirvana system (BTsN) is a medical markerless device that uses semi-immersive VR to rehabilitate patients with neurological disorders associated with motor and cognitive difficulties, even in childhood. BTsN is based on infrared optoelectronic sensors, through which the patient can interact with a virtual scenario. The system is connected to a wall or floor projector, reproducing an interactive series of exercises, using an infrared camera that analyzes the patient’s movements [11]. The previous literature on adult patients affirm the utility of this tool to enhance the functional recovery of cognition dysfunctions. In De Luca et al. [19], the use of BTsN seems to be useful in post-stroke rehabilitation, leading to improved cognitive and motor impairment with particular regard to trunk control, and visuo-spatial. Several other studies have demonstrated the usefulness of semi-immersive VR-BTsN, showing promising results regarding functional recovery and perception of quality of life in patients with various neurological diseases, including multiple sclerosis [20], traumatic brain injury [21] and Parkinson disease [22].

Thanks to its characteristic of interactive and game-like tool, active explorations are encouraged in patients and a major involvement provides motivation and enjoyment, allowing longer training sessions and improving treatment adherence. Furthermore, BTs-N has emerged as a valuable tool in the treatment of different disorders, even in pediatric age. In patients with autism spectrum disorder, for example, it was used to improve attention processes and visuospatial cognition [23]; moreover, this rehabilitation device also was used to improve balance and motor skills in children and adolescents with cerebral palsy [24].

The primary goal of this single-blind, randomized-controlled pilot study was to evaluate the feasibility of an integrated rehabilitation program with BTs-Nirvana Intervention (BTsN-I) in patients with GDD with regard to patient acceptability and sustainability along the months. The secondary aim was to evaluate the effectiveness of semi-immersive VR compared to treatment as usual (TAU).

## 2. Materials and Methods

### 2.1. Study Design

A single-blind, randomized, controlled pilot study was performed at the Child Neuropsychiatry service of the IRCCS Centro Neurolesi “Bonino Pulejo” in Messina, Italy. This study was carried out in accordance with the Declaration of Helsinki; furthermore, it was examined and approved by the Ethical Committee IRCCS Sicilia Centro Neurolesi “Bonino-Pulejo”; this clinical trial adheres to CONSORT guidelines [25], and has been registered in http://www.clinicaltrials.com (accessed on 5 June 2023) (identifier: NCT05879952). See detailed information about the CONSORT flow-chart of the study in Figure 1. Written informed consent was obtained from both caregivers or a legally authorized patient representative.

### 2.2. Inclusion Criteria and Participants

After viewing the information relating to the experimental treatment, parents remained interested in the study; a total of 50 children with GDD were screened for eligibility between December 2020 and September 2022. Inclusion criteria were (a) diagnosis of GDD, (b) age between 12 and 66 months, and (c) consistent attendance in the therapy program for the total number of sessions scheduled. Children who had other major medical conditions such as epilepsy, severe visual and auditory sensory deficits, traumatic brain injury, or other significant genetic disorders, were excluded. A total of N. 40 patients fulfilled inclusion criteria and were enrolled. The distribution of participants into experimental or control group was randomly made by a computer-generated list of arbitrary numbers, used to assign participants. Allocation was conducted by a blind researcher who did not participate in the trial.

### 2.3. Outcome Measures

Changes from pre (T0) to post (T1) interventions were evaluated by independent assessors, blind to treatment conditions. Each patient was evaluated pre- and post-treatment by the same assessor. Feasibility was assessed through service utilization analysis, which consists of engagement and the rate of participation in rehabilitation programs. To evaluate preliminary efficacy of experimental interventions, we take into account two main measures. (1) Griffiths-III Mental Development Scale (GMDS-III [26]), to track change among all developmental areas. The GMDS-III is an assessment tool for children from 0 to 72 months, that provides a General Quotient (GQ) and five different scales: Learning Bases, Language and Communication, Eye-hand Coordination, Personal-Social Emotional, and Gross-motor. For each of these areas, a score is obtained which indicates an extremely low GQ if ≤69, borderline 70–79, below average 80–89, average 90–109, above average 110–119, high 120–129, and very high > 130. (2) Clinical Global Impressions-Improvement (CGI-I [27]), to quantify and monitor patients’ progress and response to treatment, providing a clinical judgment on global improvement. In detail, the scores of the Severity section (CGI-S), ranging from 1 (normal, no disease) to 6 (seriously ill), were considered to assess the severity at baseline (T0). Moreover, the scores relating to the improvement section ranging from 1 (very improved condition) to 6 (moderately worsened).

### 2.4. Intervention

All patients underwent a cycle of 48 treatments, each lasting 45 min, in 1:1 ratio. The control group was treated with TAU twice a week, while the patients of the experimental group underwent one session of TAU and one session of BTsN per week. Therapies were carried out by qualified therapists who were randomly assigned. Each child was treated by the same therapist for the entire cycle. Both professional teams (TAU and BTsN-I teams) had similar backgrounds and professional training.

Participants assigned to control group (TAU) underwent standard neuro-psychomotor training, representatives of the existing services nationwide. During TAU sessions, patients performed exercises to promote better organization of global motor skills, improve hand-eye coordination, promote the development of language as communication, enriching representation and symbolization skills and improve the acquisition of age-appropriate developmental milestones. The treatment was tailored according to each child’s goals and preferences.

In the experimental group (BTsN-I) the conventional therapy program was integrated with the use of BTsN pediatric modules in a 1:1 ratio. BTsN treatment session included exercises designed to identify, find, chase, or move objects, with the aim of improving the perceptual-cognitive skills of each patient, through audio–visual stimuli and feedback implementing visuo-spatial skills and spatial cognition, allowing, at the same time, motor coordination and balance improvement. All the exercises had been customized according to the therapists to the individual’s treatment needs, adapting the level of difficulty to the patient’s abilities. These exercises made it possible, in a more captivating and engaging environment for the child, to work simultaneously on different cognitive and motor domains: visual perception, spatial organization, attention, memory, language, balance, posture, and coordination (See description of the games adopted during VR-based intervention in Table 1 and images in Figure 2).

### 2.5. Statistical Analysis

Data were analyzed using the R software, version 4.0.5, considering *p*-value < 0.05 as statistically significant. Due to the small sample size, a non-parametric approach was used. Thus, the Mann–Whitney U test was used to compare the two groups at baseline, whereas the Wilcoxon signed-rank test was used to compare each group between baseline and the end of the study. The Chi-squared test was used to compare proportions. Using the car package of R, for any dimension of the Griffiths III scale, an analysis of covariance (ANCOVA) was performed after testing of assumptions. The model had the test score at T1 as dependent variable, the categorical variable ‘Group’ (1 = experimental; 0 = control) as independent variable, and the outcome score at baseline (T0) as covariate. We also performed ANOVA to verify whether the model was significantly different when we fitted it including the interaction term effect “outcome score at baseline * categorical variable”.

## 3. Results

### 3.1. Baseline Characteristics of Participants

A total of N. 40 patients fulfilled inclusion criteria and were enrolled. Despite the initial agreement, n. 3 patients randomly assigned to the BTsN-I group did not start the study for personal reasons. Final sample were constituted by n. 20 subjects (13 males and 7 females, mean age 42.5 ± 13.9 months) randomly assigned to TAU and n. 17 subjects (13 males and 4 females, mean age 35.1 ± 9.8 months) to BTsN-I. A more detailed description of the two groups is in Table 2. No significant difference either for age (*p* = 0.06) or gender (*p* = 0.69) were found between experimental and control group.

### 3.2. Feasibility and Efficacy

Among the final sample, the retention rate of children who started the study and completed all the 48 sessions was 100% in both groups. The use of a semi-immersive VR instrument was well accepted by patients; there were no episodes of fear or requests to leave the room of BTsN. Moreover, no considerable alternations or side effects were observed among the BTsN-I group. Clinically, there were no significant differences between the GQ at T0. Quite the opposite, there was a statistically significant difference between the two groups at T1 regarding the GQ (t = 2.44; *p* = 0.02). As showed in Table 3, the experimental patients had significant T0–T1 differences in General Quotient Griffiths III score (*p* = 0.04), whereas emerged significant T0-T1 changes in Eye and hand coordination scale for controls (*p* < 0.01). ANCOVA results are reported in Table 4 for GQ, Foundations of Learning, Language and Communication and Eye and Hand Coordination. Assumption of homogeneity of variance was not met for Personal–Social Emotional and Gross Motor. Such results confirmed that the effect of the two treatments was significantly different in *GQ* (t = 2.44; *p* = 0.02), but also in Foundations of Learning (t = 3.66; *p* < 0.01), and ANOVA results showed a significance of the interaction term “outcome score at baseline * categorical variable” in these two models, (t = 2.49; *p* = 0.02) and (t = 3.73; *p* < 0.01), respectively. As concerning the CGI-I scale, we found a significant difference between the two groups at T1 (χ^2^ (3) = 9.16; *p* = 0.03). The distribution of raw scores related to CGI-Improvement is detailed in Figure 3.

## 4. Discussion

This single-blind, randomized-controlled pilot study investigated the feasibility and usefulness of neuro-psychomotor therapy integrated with semi-immersive VR in GDD patients, using an innovative tool, namely BTS-Nirvana. Specifically, it evaluated the engagement and the participation rate in rehabilitation programs and the effectiveness on global development improvement of semi-immersive VR compared with treatment as usual. To our knowledge, this is the first study addressing cognitive and motor rehabilitation integrated with a semi-immersive virtual reality system in children with GDD.

Few studies in the literature involve a group of GDD patients similar to ours in age distribution [28], as well as our results support the fair acceptability and feasibility of a semi-immersive virtual game in pre-school children, documented by high participant retention rates in the experimental treatment programs and the absence of adverse events. Furthermore, the results in terms of effectiveness can be interpreted as hinting at a possible utility of semi-immersive virtual reality for the rehabilitation of children with GDD. Indeed, the clinical global improvement underlined by CGI-I scale between TAU and BTsN-I groups. This finding seems well supported by significant T0-T1 differences in the General Quotient score of BTsN-I groups, with a confirmed probability of significant difference between groups (t = 2.44; *p* = 0.02), also found in the Foundations of Learning subscale (t = 3.66; *p* < 0.01).

This last specific subscale refers to a child’s ability to learn, involving different skills, such as attention and curiosity, problem solving, and processing-speed abilities. Foundations of Learning subscale also explores the ability to understand the relationships between objects or elements and the approach needed to facilitate the learning process. In early childhood, learning means exploring, recognizing similarities and differences, and understanding cause-and-effect connections. Overall, it assesses those aspects that during early childhood are the prerequisites of learning skills and that promote academic success.

Therefore, VR technology applied to neurorehabilitation treatment at preschool age can be a very valuable tool. Some reasons that would make this system advantageous could be related to the intrinsic potential of virtual reality systems. VR can offer an immersive experience to stimulate different senses at the same time; the use of auditory and visual feedback simultaneously enhances different perceptual channels, increasing awareness of one’s body actions and movements; furthermore, the presence of body shadows during VR play may improve children to a better body consciousness, too [29]. Visual observation of one’s own movements will activate the “mirror neuron system”, and the systematic exercise based on observation and imitation may improve the development of different skills in children, as shown by the studies inherent stroke rehabilitation and cerebral palsy [30,31,32]. Nonetheless, through a computer-generated virtual world, therapeutic exercises can be transformed into engaging and fun games, increasing patient compliance with treatment. Unlike other kinds of neurorehabilitation treatments, this system involves scenarios and activities that stimulate imagination and creativity within controlled and secure environments, keeping high levels of motivation and attention during the whole session [33]. Individual motivation would facilitate neuroplasticity mechanisms [34], and is important to achieve adequate compliance with therapists.

Finally, VR-based treatments give to therapists the possibility of customized settings and interventions [35]. The possibility of personalized intervention enables the transformation of each proposed scenario in a new and different environment to explore, adapting the treatment to the child, according to clinical features and personal preferences, to better targeted treatments, fostering more targeted treatments and effective interventions [36]. It is also possible to use the same scenario to work on acquiring or upgrading different skills. Specifically, BTs-N is a versatile tool capable of adapting images, sounds, and activity to the different goals of treatment.

GDD encompasses a wide range of impairments in distinct areas, such as gross and fine motor skills, speech ability, and critical aspects of learning [37]. The studies and guidelines related to treatment in GDD are still few, but the goals of therapy are very broad, encompassing all areas of development and children’s acquisition of awareness of self and others, the knowledge of strategies to copy with novelty and difficulties, the ability to plan their own behaviors and manage the possible consequences. Moreover, it is known that early and intensive interventions can improve outcomes and developmental trajectories [38,39]. Several factors have been identified to explain the greater effectiveness of early intervention: the immaturity and brain plasticity of the young child, the possibility of improving family functioning by acting on maladaptive parent–child interactions, and, lastly, the opportunity to prevent secondary complications [40]. Child development is related to learning and plasticity mechanisms generated by experience, which lead to changes in brain network and behavior. A child exposed to attractive stimuli acquires new skills, so direct experience in a challenging environment is an important source of learning [41].

Our study showed that the VR system, combined with the usual treatment, could be helpful in enhancing cognitive and learning processes, probably due to their potential for personalization, global stimulation, and engagement. Patients with GDD have complex needs, and the aim of treatment was indeed to strengthen all the areas of neurodevelopment, allowing the acquisition of praxic and perceptive, communicative-relational, symbolic, linguistic, logical-cognitive, and motor skills. Those preliminary results demonstrated that, although there were no significant changes in individual areas, the use of semi-immersive VR can be considered a valid context for global stimulation. As this is a pilot study, the sample size is limited; however, this limitation may be overcome in a future RCT. Another limitation of our study is the heterogeneity inherent in the definition of global developmental delay; GDD is a general descriptor of a broad phenotype and can result from a variety of etiological factors. However, strict exclusion criteria for serious medical conditions have been implemented to reduce this potential bias. Finally, the large age range could constitute a limit to the interpretation of the data; for this reason, we tried to keep a non-significant difference between the groups, in order to better compare the results. Nevertheless, preliminary studies allow us to test hypotheses in the investigated issue and guide future studies, and to our knowledge, this is the first study to test and confirm the feasibility and effectiveness of cognitive and motor rehabilitation integrated with a semi-immersive VR system in children with GDD.

## 5. Conclusions

Our results suggest a possible utility of VR for the rehabilitation of children with GDD, both in terms of feasibility and effectiveness. In addition, VR has been shown to be more effective than TAU in improving global perceptual-cognitive skills, probably by encouraging implicit learning through exposure to a series of fun games and targeted concrete tasks. Further studies should be promoted on larger samples to confirm these results, focusing on GDD patient with long-term follow-up.

## Figures and Tables

**Figure 1 jcm-12-04962-f001:**
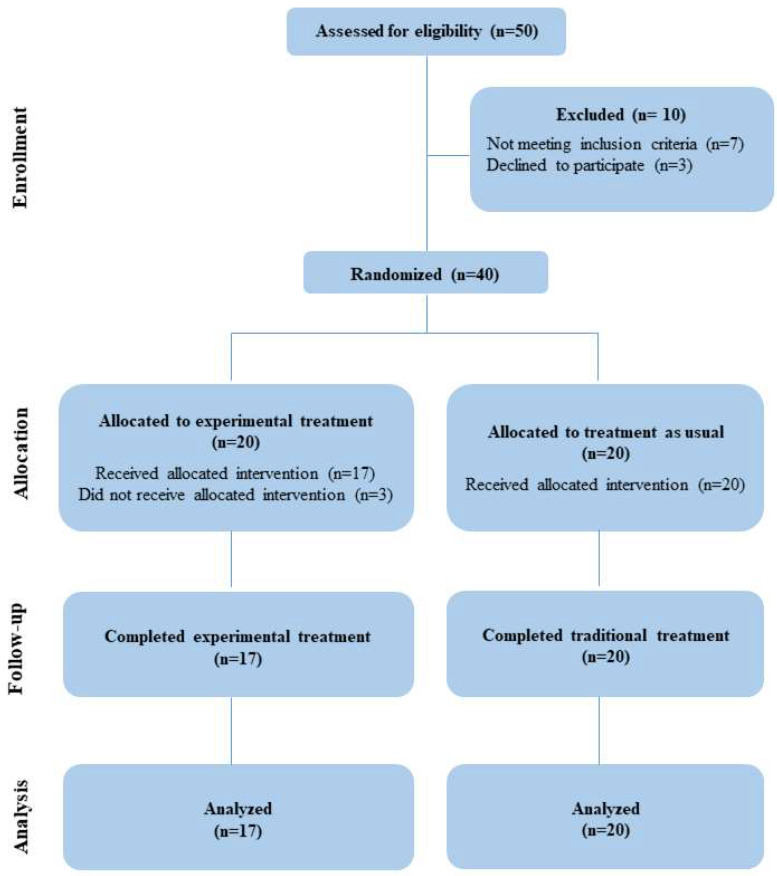
The CONSORT flow-chart with detailed information about participants in the study.

**Figure 2 jcm-12-04962-f002:**
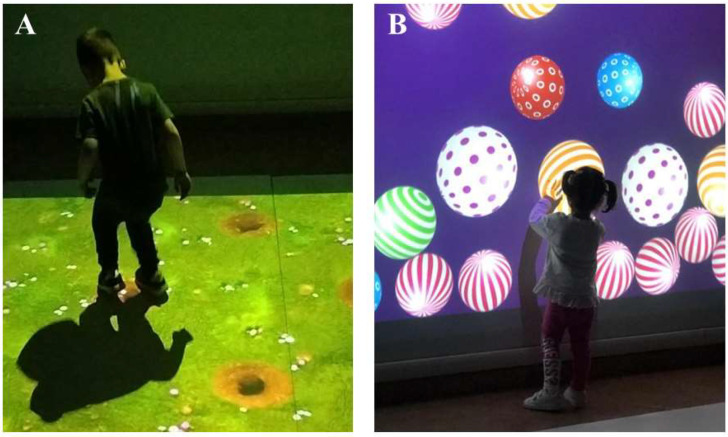
Example of two games used during BTsN intervention. Panel (**A**) Game 2—Tap Mole: the exercise is projected onto the floor and the child has to catch the moles that suddenly appear from the holes. Panel (**B**) Game 3—Balls: the exercise is projected onto the wall and the child has to bounce the balls.

**Figure 3 jcm-12-04962-f003:**
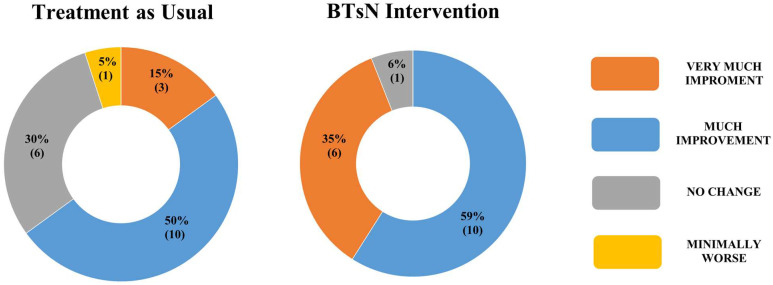
Graphic distribution of raw scores related to CGI-Improvement in children who have completed training with BTsN compared to children who have completed treatment as usual.

**Table 1 jcm-12-04962-t001:** Description of the main games used during session with BTsN by type of system projection and neuropsychological domain involved in each activity.

Scenario	Projection	Game	Neuropsychological Domain
**Balloons**	floor	To reach the balloons flying upwards and pop them with foot	Visual–motor integration, Motor coordination, Impulse control
**Tap Mole ***	floor	To capture the mole that appears randomly	Visual–spatial cognition, Motor coordination, Impulse control
**Balls ***	wall	To play with the ball to bounce it off the walls of the screen	Visual–spatial cognition, Visual–motor integration, Executive functions—planning
**Trumpets**	wall	To play the trumpets by touching them	Auditory discrimination and working memory, Visual discrimination, Executive functions—planning
**Guitar**	wall	To play the single guitar chords by hands movement	Auditory discrimination and working memory, Visual discrimination, Executive functions—planning
**Cooking**	wall	To grasp the ingredients indicated on the board and move them to the pot	Executive functions—self monitoring, organization, planning

* The scenarios of *Tap Mole* and *Balls* games of BTS-Nirvana are shown in Figure 2.

**Table 2 jcm-12-04962-t002:** Demographic characteristics of participants.

	TAU		BTsN-I		Total Sample	
	N	Mean ± SD or %	N	Mean ± SD or %	N	Mean ± SD or %
**Enrolled patients**	20		20		40	
**Final Sample**	20		17		37	
**Age in months**		35.1 ± 9.8		42.5 ± 13.9		39.1 ± 12.6
**Gender**						
*Male*	13	65.0%	13	76.5%	26	70.3%
*Female*	7	35.0%	4	23.5%	11	29.7%
*M:F Ratio*	2:1		3:1		2:1	
**CGI Severity (T0)**		4.1 ± 1		3.8 ± 1.1		4 ± 1.1

Mean ± standard deviation was used to describe continuous variables; proportions (numbers and percentages) were used to describe categorical variables.

**Table 3 jcm-12-04962-t003:** Statistical comparisons of clinical scores between baseline (T0) and follow-up (T1).

Intervention	Griffiths III	BASELINE—T0	FOLLOW-UP—T1	*p*-Value
**BTsN-I**	General Quotient	69.0 (65.0–76.0)	75.0 (61.0–87.0)	**0.04**
	Foundations of Learning	78.0 (72.0–85.0)	80.0 (74.0–87.0)	0.49
	Language and Communication	64.0 (53.0–77.0)	58.0 (50.0–78.0)	0.83
	Eye and Hand Coordination	83.0 (70.0–89.0)	80.0 (73.0–91.0)	0.28
	Personal–Social–Emotional	75.0 (69.0–88.0)	81.0 (69.0–90.0)	0.28
	Gross Motor	84.0 (69.0–96.0)	91.0 (83.0–93.0)	0.59
**TAU**	General Quotient	58.0 (49.0–77.2)	62.0 (49.0–89.0)	0.13
	Foundations of Learning	70.0 (61.5–82.0)	75.5 (60.7–91.2)	0.08
	Language and Communication	51.0 (49.0–73.2)	52.5 (49.0–83.0)	0.25
	Eye and Hand Coordination	68.5 (56.5–80.0)	76.0 (67.5–89.2)	**<0.01**
	Personal–Social–Emotional	72.5 (49.0–86.5)	70.0 (49.7–97.2)	0.21
	Gross Motor	73.0 (60.5–85.0)	75.0 (55.5–91.2)	0.48

Scores are in median (first-third quartile); significant differences are in bold. Legend: BTsN-I = BTs-Nirvana system Intervention; TAU = Treatment as usual.

**Table 4 jcm-12-04962-t004:** ANCOVA results for each covariance model.

Griffiths III	Group Coefficient				Adjusted R2
	Estimate	Std. Error	t Value	*p* Value	
**General Quotient**	35.87	14.70	2.44	**0.02**	0.56
**Foundations of Learning**	55.61	15.21	3.66	**<0.01**	0.56
**Language and Communication**	−2.95	3.31	−0.89	0.38	0.45
**Personal–Social–Emotional**	−1.36	3.17	−0.43	0.67	0.46

Significant differences between treatment effects are in bold.

## Data Availability

The data presented in this study are available on request from the corresponding author.

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
