# Peer review of "Virtual Reality Technology to Enhance Conventional Rehabilitation Program: Results of a Single-Blind, Randomized, Controlled Pilot Study in Patients with Global Developmental Delay"

_jcm, 2023, doi:10.3390/jcm12154962_

Round 1
Reviewer 1 Report
15/07/2023
Journal of Clinical Medicine
Review of the manuscript: Virtual Reality technology to enhance conventional rehabilitation program: results of a single-blind, randomized, controlled pilot study in patients with Global Developmental Delay
I appreciate the opportunity to review the manuscript – Virtual Reality technology to enhance conventional rehabilitation program: results of a single-blind, randomized, controlled pilot study in patients with Global Developmental Delay.
I find the manuscript a valuable pilot study report, which could add to the current state of knowledge on the use of virtual reality technology as a means to enhance the conventional rehabilitation program in young children with global developmental delays (GDDs).
The Authors have presented two aims. The primary aim of the study was to evaluate the feasibility of an integrated rehabilitation program with BTS-Nirvana Intervention (BTsN-I) in children with GDD with regard to patient acceptability and sustainability along the months. The secondary aim was to evaluate the effectiveness of semi-immersive virtual reality (VR) compared to treatment as usual (TAU). As such, the aims seem to be important both for the practice and future research directions. Also, the rationale for study is strongly presented in the Introduction.
The research design was appropriate to address the research questions. The researchers seek to compare the feasibility and effectiveness of traditional rehabilitation program and BTS-Nirvana Intervention, so the experimental method is justified.
Since this was a pilot study, a relatively small number of participants is acceptable. The method of randomization was appropriate. As the Authors state, “The distribution of participants into experimental or control group was randomly made by a computer-generated list of arbitrary numbers, used to assign participants. Allocation was conducted by a blind researcher who did not participated to the trial” (p. 4). The process of recruitment is quite well-described. However, there are some issues that could be explained in more detail. For example, how were families encouraged to participate in the study?
Importantly, both study group and control received the same level of care/treatment, which makes the experimental intervention well-designed.
The effects of the intervention are clearly and comprehensively reported. Definitely, the results can be applied to treatment interventions in young children.
As far as the ethics is concerned, the authors state that the “Ethical clearance was obtained from the research ethics committee at the author's institution prior to the commencement of the study. The reported procedures were carried out in compliance with the World Medical Association's Code of Ethics (Declaration of Helsinki)” (p. 7).
I find the study valuable.
The manuscript requires a proof-reading by a native speaker.
Minor problems include:
- There are some language mistakes, which in certain places make the reading difficult to follow, e.g. ‘GDDs are usually identified by caregivers, when or teachers at school raise concerns or during routine clinical evaluations by the pediatrician’ (p. 1);
- Using full terms although the abbreviation was presented (e.g. virtual reality / VR – pp. 3, 7, 9, 10);
- Some grammar (e.g. ‘did not participated’, p. 4, ‘treatments gives’, p. 9, ‘The possibility of personalized the intervention’. p. 9, ‘to confirm this results’, p. 10) and vocabulary mistakes.
Author Response
Dear editor,
We are hereby submitting the revised version of our manuscript “Virtual Reality technology to enhance conventional rehabilitation program: results of a single-blind, randomized, controlled pilot study in patients with Global Developmental Delay”, for publication as an original Article in Journal of Clinical Medicine, Special Issue “10th Anniversary of JCM—Research Updates in Developmental Psychopathology and Pediatric Neurology”.
We wish to thank the Reviewer for his helpful comments, which have each been addressed as detailed below:
I find the manuscript a valuable pilot study report, which could add to the current state of knowledge on the use of virtual reality technology as a means to enhance the conventional rehabilitation program in young children with global developmental delays (GDDs). The Authors have presented two aims. The primary aim of the study was to evaluate the feasibility of an integrated rehabilitation program with BTS-Nirvana Intervention (BTsN-I) in children with GDD with regard to patient acceptability and sustainability along the months. The secondary aim was to evaluate the effectiveness of semi-immersive virtual reality (VR) compared to treatment as usual (TAU). As such, the aims seem to be important both for the practice and future research directions. Also, the rationale for study is strongly presented in the Introduction. The research design was appropriate to address the research questions. The researchers seek to compare the feasibility and effectiveness of traditional rehabilitation program and BTS-Nirvana Intervention, so the experimental method is justified. Since this was a pilot study, a relatively small number of participants is acceptable. The method of randomization was appropriate. As the Authors state, “The distribution of participants into experimental or control group was randomly made by a computer-generated list of arbitrary numbers, used to assign participants. Allocation was conducted by a blind researcher who did not participated to the trial” (p. 4). The process of recruitment is quite well-described. However, there are some issues that could be explained in more detail. For example, how were families encouraged to participate in the study?
We thank Reviewer 1 for expressing appreciation for our work and for underscoring its relevance to the field. We have better explained the process of children recruitment and families’ involvement in lines 127-129.
Minor problems include:
- There are some language mistakes, which in certain places make the reading difficult to follow, e.g. ‘GDDs are usually identified by caregivers, when or teachers at school raise concerns or during routine clinical evaluations by the pediatrician’ (p. 1);
We thank Reviewer 1 for pointing out this mistake, which has been corrected (line 38).
- Using full terms although the abbreviation was presented (e.g. virtual reality / VR – pp. 3, 7, 9, 10);
We thank Reviewer 1 for pointing out this mistake, we have replaced all full terms with abbreviations in the text.
- Some grammar (e.g. ‘did not participated’, p. 4, ‘treatments gives’, p. 9, ‘The possibility of personalized the intervention’. p. 9, ‘to confirm this results’, p. 10) and vocabulary mistakes.
We performed a final revision by a native speaker to correct linguistic errors.
We thank again the Reviewers for their helpful comments, which have each been addressed in this revised and much-improved version of our manuscript. We sincerely hope that you will now deem this contribution acceptable for publication in Journal of Clinical Medicine.
Looking forward to the final outcome of the review process, I send you my best regards.
Sincerely,
Dr. Carmela Settimo

Reviewer 2 Report
Title: Virtual Reality technology to enhance conventional rehabilitation program: results of a single-blind, randomized, controlled pilot study in patients with Global Developmental Delay
This single-blind, randomized, controlled pilot study aimed to evaluate the feasibility and effectiveness of conventional rehabilitation programs integrated with the BTs-Nirvana virtual reality system compared to only usual treatment in patients with global developmental delay.
Main comments
In general, the manuscript is well-written. Be careful with the spaces and points in the text and DOI underlined and bibliography norms in the references. Some specific comments are presented below.
Specific comments
0. Abstract
- Line 25: Include (GQ) after “General Quotient”.
- Lines 30-31: “Global developmental delay”, “rehabilitation treatment” and “interactive learning environments” Keywords are not MeSH terms. Write instead of those words “Neurodevelopmental disorder”, “rehabilitation” and “interactive learning”. It would be appropriate to use 4-6 MeSH terms in the Keywords section.
1. Introduction
- Line 94: In line 234, you refer that this is the first single-blind, randomized, controlled pilot study that used semi-immersive virtual reality as treatment in children with Global Developmental Delay. However, it would be interesting to add examples of specifically BTS-Nirvana system as treatment in other neurological diseases (adults or children).
2. Materials and Methods
- Figure 1: Review the number of participants in enrollment (assessed for eligibility and excluded) before the randomization.
- Line 178: You must explain the statistical analysis for sample normal distribution and the reasons why Mann-Whitney U and Wilcoxon tests were used.
3. Results
- No comments.
4. Discussion
- Lines 279-282: Delete these lines and include them after line 94 as examples of specifically BTS-Nirvana system as treatment in children neurological diseases.
- Line 306: Write more limitations of the study and not only the reduced sample size (for instance, the wide range of months or the several pathologies that involves global developmental delay).
5. Conclusions
- Lines 310-315: It would be appropriate to end with a “Conclusions” paragraph related to the objectives of the essay. You do not mention the secondary aim in these lines.
References
- Review DOI underlined and bibliography norms.
Author Response
Dear editor,
We are hereby submitting the revised version of our manuscript “Virtual Reality technology to enhance conventional rehabilitation program: results of a single-blind, randomized, controlled pilot study in patients with Global Developmental Delay”, for publication as an original Article in Journal of Clinical Medicine, Special Issue “10th Anniversary of JCM—Research Updates in Developmental Psychopathology and Pediatric Neurology”.
We wish to thank the Reviewer for his helpful comments, which have each been addressed as detailed below:
Reviewer 2
This single-blind, randomized, controlled pilot study aimed to evaluate the feasibility and effectiveness of conventional rehabilitation programs integrated with the BTs-Nirvana virtual reality system compared to only usual treatment in patients with global developmental delay.
Main comments
In general, the manuscript is well-written. Be careful with the spaces and points in the text and DOI underlined and bibliography norms in the references. Some specific comments are presented below.
We wish to thank Reviewer 2 for expressing appreciation for our work. His/her comment has been addressed by arranging punctuation and references according to editorial standards. Below we list point by point all the changes made as required.
Specific comments
- Abstract
- Line 25: Include (GQ) after “General Quotient”.
Both in the Abstract and throughout the manuscript we have rechecked and corrected the use of abbreviations/full terms.
- Lines 30-31: “Global developmental delay”, “rehabilitation treatment” and “interactive learning environments” Keywords are not MeSH terms. Write instead of those words “Neurodevelopmental disorder”, “rehabilitation” and “interactive learning”. It would be appropriate to use 4-6 MeSH terms in the Keywords section.
We thank the reviewer for this correction, we followed his suggestions and changed all keywords as requested.
- Introduction
- Line 94: In line 234, you refer that this is the first single-blind, randomized, controlled pilot study that used semi-immersive virtual reality as treatment in children with Global Developmental Delay. However, it would be interesting to add examples of specifically BTS-Nirvana system as treatment in other neurological diseases (adults or children).
As suggested, we include Lines 279-282 as examples of specifically BTS-Nirvana system in Introduction paragraph; furthermore, we have added several examples of interventions with the BTS-Nirvana system in adult or pediatric patients with neurological diseases in line 91-106.
- Materials and Methods
- Figure 1: Review the number of participants in enrollment (assessed for eligibility and excluded) before the randomization.
Thanks to the reviewer for the correction, Figure 1 has been revised and corrected.
- Line 178: You must explain the statistical analysis for sample normal distribution and the reasons why Mann-Whitney U and Wilcoxon tests were used.
We explained in more detail the analysis performed, as now stated in the Materials and Methods (lines 191-195): “Data were analyzed using the R software - version 4.0.5, considering p-value < 0.05 as statistically significant. Due to the small sample size, a nonparametric approach was used. Thus, the Mann–Whitney U test was used to compare the two groups at baseline, whereas the Wilcoxon signed-rank test to compare each group between baseline and the end of the study. The Chi-squared test was used to compare proportions”.
- Results
- No comments.
- Discussion
- Lines 279-282: Delete these lines and include them after line 94 as examples of specifically BTS-Nirvana system as treatment in children neurological diseases.
This correction has been made, see indications inherent in the paragraph “Introduction”.
- Line 306: Write more limitations of the study and not only the reduced sample size (for instance, the wide range of months or the several pathologies that involves global developmental delay).
We tried to expand the limitations section; this is now stated in lines 312-319.
- Conclusions
- Lines 310-315: It would be appropriate to end with a “Conclusions” paragraph related to the objectives of the essay. You do not mention the secondary aim in these lines.
We thank the reviewer for this suggestion, we have added a paragraph dedicated to the conclusions (Line 325), attending to both aims of this study.
References
- Review DOI underlined and bibliography norms.
We revised references according to bibliography norms.
We thank again the Reviewers for their helpful comments, which have each been addressed in this revised and much-improved version of our manuscript. We sincerely hope that you will now deem this contribution acceptable for publication in Journal of Clinical Medicine.
Looking forward to the final outcome of the review process, I send you my best regards.
Sincerely,
Dr. Carmela Settimo